# Sialic acid *O*-acetylation patterns and glycosidic linkage type determination by ion mobility-mass spectrometry

Gaël M. Vos [1,5], Kevin C. Hooijschuur[1,5], Zeshi Li [1], John Fjeldsted[2], Christian Klein[2], Robert P. de Vries[1], Javier Sastre Toraño[1] ✉ & Geert-Jan Boons [1,3,4] ✉

*O*-acetylation is a common modification of sialic acids that has been implicated in a multitude of biological and disease processes. A lack of analytical methods that can determine exact structures of sialic acid variants is a hurdle to determine roles of distinct *O*-acetylated sialosides. Here, we describe a drift tube ion mobility-mass spectrometry approach that can elucidate exact *O*-acetylation patterns as well as glycosidic linkage types of sialosides isolated from complex biological samples. It is based on the use of a library of synthetic *O*-acetylated sialosides to establish intrinsic collision cross section (CCS) values of diagnostic fragment ions. The CCS values were used to characterize *O*-acetylated sialosides from mucins and *N*-linked glycans from biologicals as well as equine tracheal and nasal tissues. It uncovered contrasting sialic acid linkage types of acetylated and non-acetylated sialic acids and provided a rationale for sialic acid binding preferences of equine H7 influenza A viruses.

Sialic acids are negatively charged nine-carbon monosaccharides that are often part of complex glycans of higher animals[1,2]. Several pathogenic microorganisms also express sialylated glycoconjugates which are used for molecular mimicry to evade host immune detection[3–5]. Sialoglycans regulate many biological and disease processes[6–8], and can function as receptor for many pathogens including viruses, bacteria and protozoa[9]. Sialylation of glycans also modulates half-life, immunogenicity, and properties of biologicals[10–12].

*N*-Acetylneuraminic acid (Neu5Ac) and *N*-glycolylneuraminic acid (Neu5Gc) are major forms of sialic acid expressed by mammals. They can be modified by acetyl esters at the 4-, 7-, 8-, and/or 9-position to give as many as 15 different patterns of *O*-acetylation[13]. Further structural diversity comes from different Neu5Ac and Neu5Gc glycosidic linkage types and the most common ones are α2,3-linked to galactose (Gal), α2,6-linked to Gal and *N*-acetyl-galactosamine (GalNAc) and α2,8-linked to another sialic acid. The resulting glycotopes can be presented at different underlying glycan moieties and can be part of

Asn- and Ser/Thr-linked glycans (*N*- and *O*-linked glycans, respectively), glycolipids as well as free floating glycans such as human milk oligosaccharides. The expression of sialic acids is regulated in a developmental and tissue-specific manner. Furthermore, there are marked differences in the expression of sialoglycans in different species, which likely is due to evolutionary pressure evoked by host–pathogen interactions[14].

A growing body of literature associates sialic acid *O*-acetylation with various diseases including cancer, immune disorders, and infection[15–18]. *O*-acetylation can preclude recognition by glycan binding proteins such as the Siglec immune-receptors and complement protein factor H and as a result can function as a molecular switch[19]. It also substantially reduces the rate of hydrolysis by several human endogenous sialidases thereby regulating properties of glycoconjugates such as turnover and degradation[20]. In addition, it can block the activity of bacterial sialidases and by this means protect the integrity of the epithelial mucus barrier. *O*-acetylated sialic acids serve also as

[1]Department of Chemical Biology and Drug Discovery, Utrecht Institute for Pharmaceutical Sciences, Utrecht University, Universiteitsweg 99, 3584 CG Utrecht, The Netherlands. [2]Agilent Technologies, Santa Clara, CA 95051, USA. [3]Bijvoet Center for Biomolecular Research, Utrecht University, 3584 CG Utrecht, The Netherlands. [4]Complex Carbohydrate Research Center and Department of Chemistry, University of Georgia, 315 Riverbend Road, Athens, GA 30602, USA. [5]These authors contributed equally: Gaël M. Vos, Kevin C. Hooijschuur. ✉e-mail: j.sastretorano@uu.nl; g.j.p.h.boons@uu.nl

receptors for many viruses including embecoviruses (family Coronaviridae), toroviruses (Tobaniviridae), and influenza C and D viruses (Orthomyxoviridae)[21]. Recent analysis of receptor specificities of these viruses infecting different species of animals and humans demonstrated host-specific patterns of receptor recognition in relationship to both the pattern of acetylation and glycosidic linkage type. It was found that human respiratory viruses uniquely bind 9-O-acetylated α2,8-linked disialoside found on glycosphingolipids[21].

Despite advances, the roles of distinct O-acetylated sialosides in health and disease remain difficult to explore. A major hurdle is a lack of convenient experimental approaches to determine exact structures of the sialic acid variants including the pattern of acetylation and glycosidic linkage type. These molecules are chemically labile and prone to acetyl ester migration and hydrolysis complicating isolation and characterization[22]. O-acetylated sialosides have been analyzed at the peptide level[23] or released from glycoconjugates by treatment with acetic acid at elevated temperatures, labeled with 1,2-dihydroxy-4,5-methylendioxybenzol (DMB) and then analyzed by high-performance liquid chromatography (HPLC) or liquid chromatography-mass spectrometry (LC-MS)[24]. Although these methods are sensitive, they do not provide information about glycosidic linkage type. In another approach, carboxylic acids of sialosides of N-linked glycan were modified as methylamines to increase the stability and then analyzed by matrix-assisted laser desorption/ionization time-of-flight (MALDI-TOF) MS[25]. This approach only provides compositions and cannot assign positions of acetyl esters and glycosidic linkage type. Soluble hemagglutinin-esterases have been used as lectins for tissue staining to qualitative assess acetyl ester display[26–28]. Although powerful, virolectins exhibit promiscuous binding behavior and cannot detect all common O-acetylation patterns[21].

Here, we describe a drift tube ion-mobility (IM)-MS approach that can elucidate exact O-acetylation patterns as well as glycosidic linkage types of sialosides isolated from complex biological samples. In IM spectrometry (IMS), gas-phase ions are separated based on their mobility through a gas-filled drift cell under the influence of an electric field. The mobility of the ions depends on their charge state and size as well as on their shape, making IMS suitable for the separation of isomers[29,30]. In drift tube IMS, large ions experience more ion-neutral collisions and migrate through the drift cell at a lower speed than small ions, while ions with a higher charge state migrate faster than ions with a lower charge state, resulting in a distinctive arrival time distribution (ATD) at the end of the drift cell. The arrival times can be converted into rotationally averaged ion-neutral collision cross sections (CCS), as described by the fundamental Mason-Schamp equation[29,31]. These intrinsic CCS values are related to the surface areas of the ions and provide, in combination with m/z values, molecular descriptors for reliable compound identification.

Isomeric glycans can have different surface areas and therefore may exhibit distinct CCS values[32–34]. Several studies have shown that contemporary IM-MS equipment offers sufficient resolution to separate isomeric glycans[30,35] and has the potential to determine exact structures. Trapped IM-MS, for example, has been applied in combination with electronic excitation dissociation to separate several glycan fragments[36]. Drift tube and traveling wave (TW)IM-MS have been used to separate glycan conformers[34] and fragment ions of sialic acid isomers[37], respectively, to determine sialic acid linkages of released glycans. Cyclic TWIM-MS has very high resolution capabilities and has been employed to resolve anomers and open-ring forms of oligosaccharides[38]. In addition, the TWIM technique has been used in structures for lossless ion manipulation (SLIM)-based IMS to achieve very high resolution separation of isomeric glycans[39] and exact glycan structure elucidation in combination with cryogenic infrared spectroscopy-MS[40]. Despite these advances, the challenge of implementing IM-MS for exact glycan structure determination is a lack of standards to determine CCS values of diagnostic fragment ions.

In this study, we employed a large panel of synthetic O-acetylated N-acetyl and N-glycolylneuraminic acids to establish CCS values of diagnostic B₁ and B₃ fragment ions[21]. We demonstrate that the library of CCS values can be employed to determine the exact pattern of O-acetylation and glycosidic linkage type of sialosides isolated from complex biological samples. An important aspect of the approach was the implementation of methods that can release N- and O-glycans without affecting acetyl esters. N-linked glycans could be released by treatment with PNGase or ENDO-F2 under neutral or slightly acidic conditions whereas O-glycans could be oxidatively cleaved by sodium hypochlorite at pH 6.8[41]. The approach can be employed for the analysis of biotherapeutics for quality control and be used to examine structures of N- and O-linked sialosides obtained from tissue samples and secreted mucins. The uncovering of sialylation provides insights in the biosynthesis of this class of compounds. Furthermore, analysis of equine upper airway tissue uncovered contrasting sialic acid linkage types of acetylated and non-acetylated sialic acids which provided a rationale for sialic acid binding preferences of equine H7 influenza A viruses.

## Results
### IM-MS of O-acetylated sialosides

Previously, we developed a methodology to synthesize sialosides that differ in the pattern of O-acetylation and glycosidic linkage type[21]. It is based on the chemical synthesis of 2,3-, 2,6- and 2,8-linked sialoglycans having acetyl esters at C-4, C-7, and C-9 (Fig. 1a). These compounds were treated with hemagglutinin-esterases (HE) from bovine coronavirus (BCoV) or mouse hepatitis virus strain S (MHV-S), which cleave acetyl esters on the C-9 and C-4 position of sialic acid, respectively and in combination with controlled acetyl ester migration from C-7 to C-9 could readily be diversified to give a large panel of compounds (Fig. 1b, compounds 1–27). In this study, the resulting sialoglycans were employed as standards to establish a library of CCS values of informative fragment ions for compound identification. Each compound was analyzed by HPLC coupled to an Agilent Technologies 6560B drift tube IM/quadrupole time-of-flight MS. Measurements were performed in positive ion mode and in-source collisionally activated dissociation was used by applying a fragmentor voltage of 600 V, to achieve glycosidic bond fragmentation and produce informative B-ions (Fig. 1c). The fragment ions were analyzed by drift tube IM-MS using nitrogen as buffer gas. Arrival time distributions (ATDs) of standards were obtained in triplicate using drift tube IM with ion multiplexing and associated demultiplexing[42] combined with high-resolution demultiplexing[43]. CCS values of the ions were directly calculated from their IM arrival times (Fig. 2 and Table 1) using single field CCS calibration with standards with known m/z and CCS values.

Most fragment ions gave rise to a unimodal ATD, while a few distributions showed more complex signals for a singular ion, which is most likely due to the presence of different gas phase conformers[44]. B₁ ions of the three mono-acetylated Neu5Ac types could be resolved with the fragment ions from 4-O-acetylated Neu5Ac having the smallest CCS, followed by the fragment ions of 7- and 9-O-acetylated Neu5Ac. The B₁ ions for the Neu5Gc derivatives showed a similar trend although the B₁ ions from 4- and 7-O-acetylated Neu5Gc had a smaller difference in CCS value. B₁ ions could also distinguish 4,9- and 7,9-di-O-acetylated Neu5Ac and Neu5Gc. In the case of Neu5Ac the fragment ion from the 7,9-isomer was the smaller isomer whereas the opposite was observed for Neu5Gc with the fragment ion from the 4,9-isomer having the smaller CCS. As anticipated, the CCS values of B₁ ions derived from α2,3- and α2,6-linked sialosides, having the same acetylation pattern, as well as those from α2,8-linked Neu5Ac derivatives 25–27 were in close agreement.

B₃ ions also provide informative structural information, and it was found that those of the α2,3-linked structures have longer drift times and thus higher CCS values compared to the corresponding α2,6-

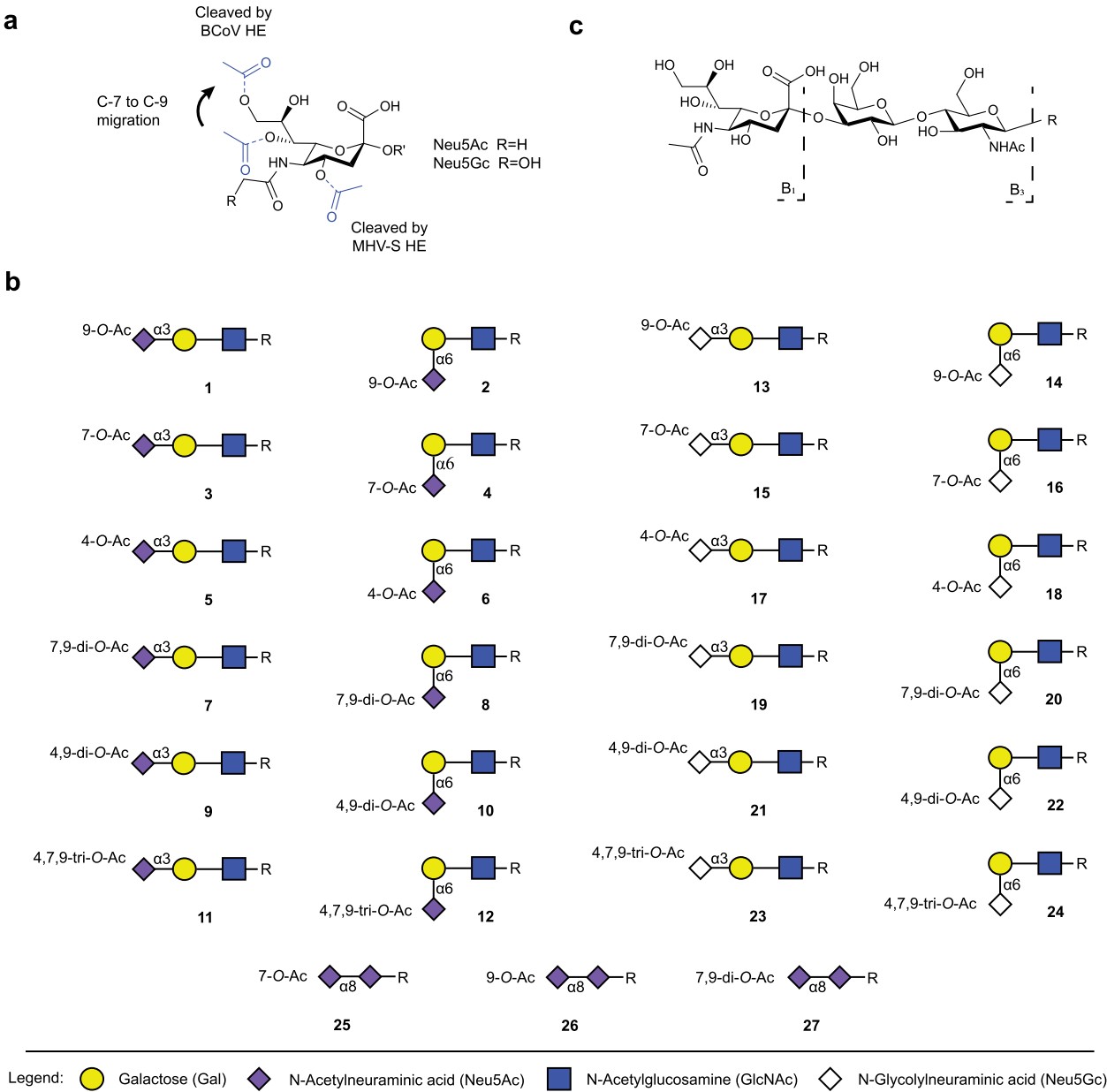

**Fig. 1 | O-acetylated sialoglycan standards. a** Structure of O-acetylated Neu5Ac and Neu5Gc. Annotated to the structure are the reactions that enable the controlled synthesis of well-defined O-acetylation patterns. **b** Library of synthetic glycan standards containing O-Acetylated (O-Ac) Neu5Ac (**1–12**), Neu5Gc (**13–24**) and di-Neu5Ac (**25–27**); R = pentylaminobiotin. **c** Nomenclature of glycan fragment ions[54].

linked compounds, making it possible to determine sialic acid glycosidic linkage type. Furthermore, most B₃ ions having the same number of acetyl esters could be separated enabling determination of O-acetylation positions. Only the B₃ ions derived from the 7- and 9-O-acetylated α2,6-linked Neu5Ac-LacNAc (**2** and **4**) and Neu5Gc-LacNAc (**14** and **16**) could not be sufficiently resolved for direct unambiguous O-acetyl position determination. De-O-acetylation, which can result in misassignment of structures was minimally observed (<1%).

## O-acetylation of N-linked sialosides derived from biologicals

The database of ATDs and CCS values was employed to analyze O-acetylated sialosides of two biologicals that are produced in Chinese hamster ovary (CHO) cell lines. These cell lines are commonly employed for the expression of therapeutic proteins and have been reported to modify N-linked glycans with O-acetylated sialosides[45]. The pattern of O-acetylation can, however, differ between CHO cell lines

and change during purification or storage[46]. Myozyme (Genzyme) is an alpha-glucosidase employed for enzyme replacement therapy for patients with Pompe disease and has at least 6 confirmed N-glycosylation sites. Both mono- and di-O-acetylated sialic acids have been reported on the N-glycans of Myozyme[47]. Aflibercept (Regeneron) is a fusion protein of the extracellular domain of the human VEGF receptor modified and the Fc region of human IgG1 and used for the treatment of metastatic colorectal cancer. Aflibercept has four N-glycosylation sites on the VEGF receptor domain and one on its Fc domain which predominantly carry biantennary N-glycans[48]. The recombinant glycoproteins were dialyzed to remove additives and then the N-glycans were released enzymatically by treatment with ENDO-F2 under mild acidic conditions (100 mM sodium acetate, pH 4.5) to prevent acetyl ester migration and hydrolysis. Pretreatment of the glycoprotein with ENDO-F2 releases the abundant biantennary complex N-glycans by cleavage between the two GlcNac residues of the chitobiose core,

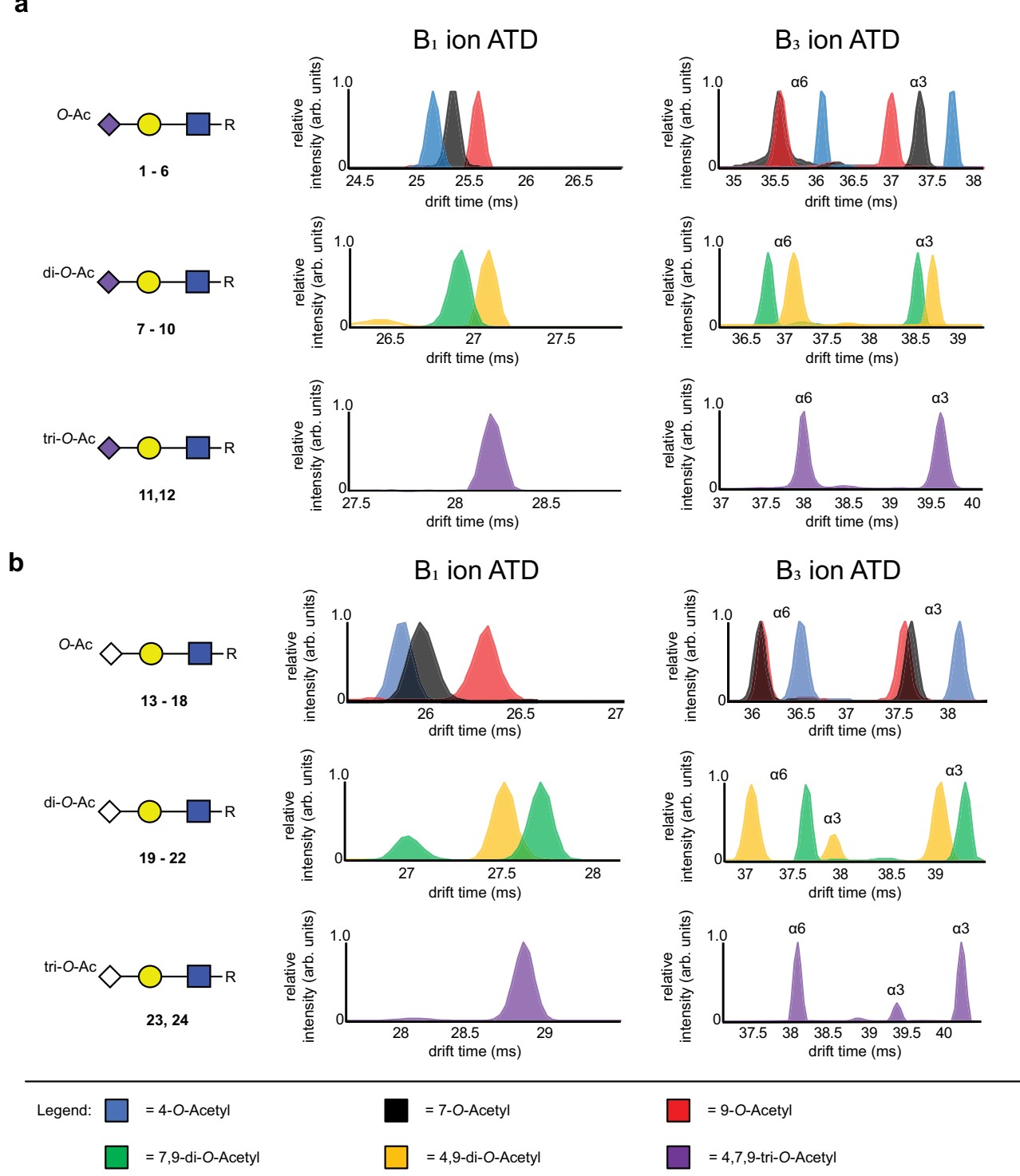

**Fig. 2 | Arrival time distributions of sialoglycan fragment ions. a** Combined arrival time distributions (ATDs) of single charged $B_1$ and $B_3$ fragment ions of mono- ($B_1$ at m/z 334.1134, $B_3$ at m/z 699.2455), di- ($B_1$ at m/z 376.1243, $B_3$ at m/z 741.2560) and tri-O-Acetylated (O-Ac) Neu5Ac ($B_1$ at m/z 418.1348, $B_3$ at m/z 783.2666). **b** Combined ATDs of single charged $B_1$ and $B_3$ fragment ions of mono- ($B_1$ at m/z 350.1090, $B_3$ at m/z 715.2404), di- ($B_1$ at m/z 392.1192, $B_3$ at m/z 757.2509) and tri-O-Acetylated Neu5Gc ($B_1$ at ᵃm/z 434.1293, $B_3$ at m/z 799.2615).

which should improve the detection of low abundant O-acetylated structures that may otherwise not be detected due the heterogeneity introduced by core fucosylation. A total of 29 biantennary N-glycans were identified for Myozyme and 21 for Aflibercept (Supplementary Data 1 and 2). The MS data for Myozyme revealed the presence of O-acetyl modified sialosides on ~3% of the total N-glycan content, which corresponds to ~15% of the sialylated N-glycans. For five

N-glycans, the measured molecular weights indicated the presence of mono- and di-O-acetylated sialic acids (Fig. 3b). The CCS of the $B_1$ fragment ions confirmed the presence of 7- and 9-O-mono- and 7,9-di-O-acetylated Neu5Ac. $B_3$ fragment ions could be detected for most of the mono-O-acetylated N-glycans and confirmed these are α2,3-linked sialosides that are acetylated at the C-7- or C-9 position. The CCS of $B_1$ fragment ions detected for three di-O-acetylated N-glycans correlated

**Table 1 | Average $^{DT}CCS_{N2}$ values of $B_1$ and $B_3$ ions of compounds 1–24 ($n$ = 3)**

| Compound | Sialic acid | Sialic acid linkage | Acetyl ester position | $m/z$ $B_1$-ion | $m/z$ $B_3$-ion | CCS $B_1$-ion (Å$^2$) | CCS $B_3$-ion (Å$^2$) |
|---|---|---|---|---|---|---|---|
| 1 | Neu5Ac | α2,3 | 9 | 334.1134 | 699.2450 | 174.77(±0.05) | 248.26(±0.08) |
| 2 | Neu5Ac | α2,6 | 9 | 334.1134 | 699.2450 | 174.82(±0.06) | 238.48(±0.04) |
| 3 | Neu5Ac | α2,3 | 7 | 334.1134 | 699.2450 | 173.34(±0.02) | 250.74(±0.07) |
| 4 | Neu5Ac | α2,6 | 7 | 334.1134 | 699.2450 | 173.23(±0.04) | 238.38(±0.11) |
| 5 | Neu5Ac | α2,3 | 4 | 334.1134 | 699.2450 | 171.90(±0.01) | 253.59(±0.02) |
| 6 | Neu5Ac | α2,6 | 4 | 334.1134 | 699.2450 | 171.91(±0.08) | 242.24(±0.15) |
| 7 | Neu5Ac | α2,3 | 7,9 | 376.1243 | 741.2561 | 183.53(±0.07) | 258.75(±0.06) |
| 8 | Neu5Ac | α2,6 | 7,9 | 376.1243 | 741.2561 | 183.36(±0.02) | 246.26(±0.08) |
| 9 | Neu5Ac | α2,3 | 4,9 | 376.1243 | 741.2561 | 184.48(±0.04) | 259.88(±0.11) |
| 10 | Neu5Ac | α2,6 | 4,9 | 376.1243 | 741.2561 | 184.29(±0.03) | 248.73(±0.08) |
| 11 | Neu5Ac | α2,3 | 4,7,9 | 418.1348 | 783.2663 | 191.52(±0.05) | 265.83(±0.04) |
| 12 | Neu5Ac | α2,6 | 4,7,9 | 418.1348 | 783.2663 | 191.52(±0.06) | 254.13(±0.02) |
| 13 | Neu5Gc | α2,3 | 9 | 350.1090 | 715.2401 | 179.30(±0.11) | 253.37(±0.03) |
| 14 | Neu5Gc | α2,6 | 9 | 350.1090 | 715.2401 | 179.30(±0.05) | 242.17(±0.12) |
| 15 | Neu5Gc | α2,3 | 7 | 350.1090 | 715.2401 | 176.69(±0.11) | 253.37(±0.03) |
| 16 | Neu5Gc | α2,6 | 7 | 350.1090 | 715.2401 | 176.89(±0.06) | 241.62(±0.07) |
| 17 | Neu5Gc | α2,3 | 4 | 350.1090 | 715.2401 | 176.30(±0.05) | 257.20(±0.03) |
| 18 | Neu5Gc | α2,6 | 4 | 350.1090 | 715.2401 | 176.42(±0.23) | 244.93(±0.01) |
| 19 | Neu5Gc | α2,3 | 7,9 | 392.1192 | 757.2510 | 187.02(±0.03) | 261.56(±0.09) |
| 20 | Neu5Gc | α2,6 | 7,9 | 392.1192 | 757.2510 | 186.98(±0.07) | 247.88(±0.02) |
| 21 | Neu5Gc | α2,3 | 4,9 | 392.1192 | 757.2510 | 188.31(±0.06) | 263.43(±0.09) |
| 22 | Neu5Gc | α2,6 | 4,9 | 392.1192 | 757.2510 | 188.25(±0.05) | 251.71(±0.05) |
| 23 | Neu5Gc | α2,3 | 4,7,9 | 434.1293 | 799.2594 | 195.12(±0.11) | 268.92(±0.02) |
| 24 | Neu5Gc | α2,6 | 4,7,9 | 434.1293 | 799.2594 | 195.22(±0.07) | 254.64(±0.06) |

with a 7,9-di-$O$-acetylation pattern in our database. CCS values of $B_3$ fragment ions could be determined for the two most abundant structures and confirmed α2,3-linkage types. These observations agree with the fact that CHO cells only express α2,3-linked sialosides[49].

The MS data showed only one $O$-acetylated $N$-glycan in the Aflibercept sample. Ion mobility analysis of the $B_1$ ion revealed that this $N$-glycan contains exclusively 9-$O$-acetylated sialic acid (Fig. 3a). Due to the low abundancy, no $B_3$ fragment ions could be detected and therefore the sialic acid linkage type could not be established. Collectively, the results demonstrate that IM-MS can assign both acetyl ester position and sialic acid linkage type of $N$-glycans released from biologicals using CCS values of $B_1$ and $B_3$ fragment ions. Previously, only compositions of $O$-acetylated structures could be determined by MS showing the presence of mono- and di-$O$-acetylated sialosides in Myozyme[47].

### $O$-acetylation of $N$-linked sialosides from equine tracheal and nasal tissues

Respiratory viruses, which cause enormous disease burden, commonly employ glycans for cell attachment and/or entry[50]. The relentless pressure of microbial infections at the mucosal interface has driven the evolution of host and pathogen[51]. It has shaped the glycomes of the host and even closely related species express substantially different glycans. In turn, pathogens evolved glycan receptor specificities that determine host range and tissue tropism. To understand this co-evolution, it is necessary to determine exact structures of glycans of respiratory tissues. Previous tissue staining of equine respiratory tract tissues with the plant lectins SNA and MAH revealed similar levels of α2,3- and α2,6-linked sialic acids[52]. Furthermore, the 4-$O$-acetyl specific lectin MHV-S showed abundant presence of this sialoglycan[52]. Although MHV-S has an obligatory requirement for sialate-4-$O$-acetylation, it tolerates the presence of acetyl esters at C-7 and C-9 and can bind to 2,3- as well as 2,6-linked sialosides. Thus, this lectin cannot determine fine structural details of $O$-acetylated sialosides. Therefore,

we analyzed $N$-glycans obtained from equine tracheal and nasal tissues by the IM-MS approach.

$N$-glycans from nasal tissue from three different horses were released using PNGaseF in TRIS buffer (100 mM, pH 7.0) and analyzed by IM-MS. The MS data of the nasal samples revealed the presence of mainly biantennary glycans. The fragmentation spectra, obtained after in-source activation, showed several $N$-glycan subclasses in addition to the terminal sialylated structures, such as core fucosylated and α-galactosylated structures. Core fucosylation was demonstrated by the presence of $Y_{1α}$ and $Y_2$ fragment ions with $m/z$ 587.3286 and $m/z$ 790.4080 (Supplementary Fig. 1), while antenna fucosylation could be excluded by the absence of fucosyl-LacNAc fragment ions with $m/z$ 512.1974. α-Galactosylation was identified by a Hex$_2$HexNAc fragment ($m/z$ 528.1923) arising in high abundance from a single cleavage resulting in a Gal$_2$GlcNAc $B_3$ fragment ion (Supplementary Fig. 1). No further diagnostic fragment ions were detected in the samples for exact glycan structure determination, although the identification of specific fragment ions in combination with known biosynthetic pathways[53] allowed for the compilation of a list of glycan compositions (Supplementary Data 3–5 and Supplementary Note 1). The eight glycan compositions that were shared across all three tissue samples were the most abundant ones (adding up to over 60% for each sample, Supplementary Fig. 2). $O$-acetylation was observed on 33–53% (relative abundance) of the total $N$-glycan content of the three samples (Supplementary Data 3–5). Two di-sialylated $N$-glycans with one or two acetyl esters accounted for 73% of the observed $O$-acetylated ions in one sample and over 98% in the other samples.

$O$-acetylated sialosides were identified by their $B_1$ and $B_3$ fragment ions by IM as depicted in Fig. 4a, b. According to the CCS values of the $B_1$ fragment ions, all $O$-acetylated sialic acids are exclusively 4-$O$-acetylated. CCS values of the $B_3$ ions confirmed the $O$-acetylation assignment and revealed the presence of both α2,3- and α2,6-linked Neu5Ac and Neu5Gc derivatives. $B_3$ ions corresponding to α2,6-linked-Neu4,5di-Ac were abundantly present, whereas those for α2,3-linked Neu4,5di-Ac

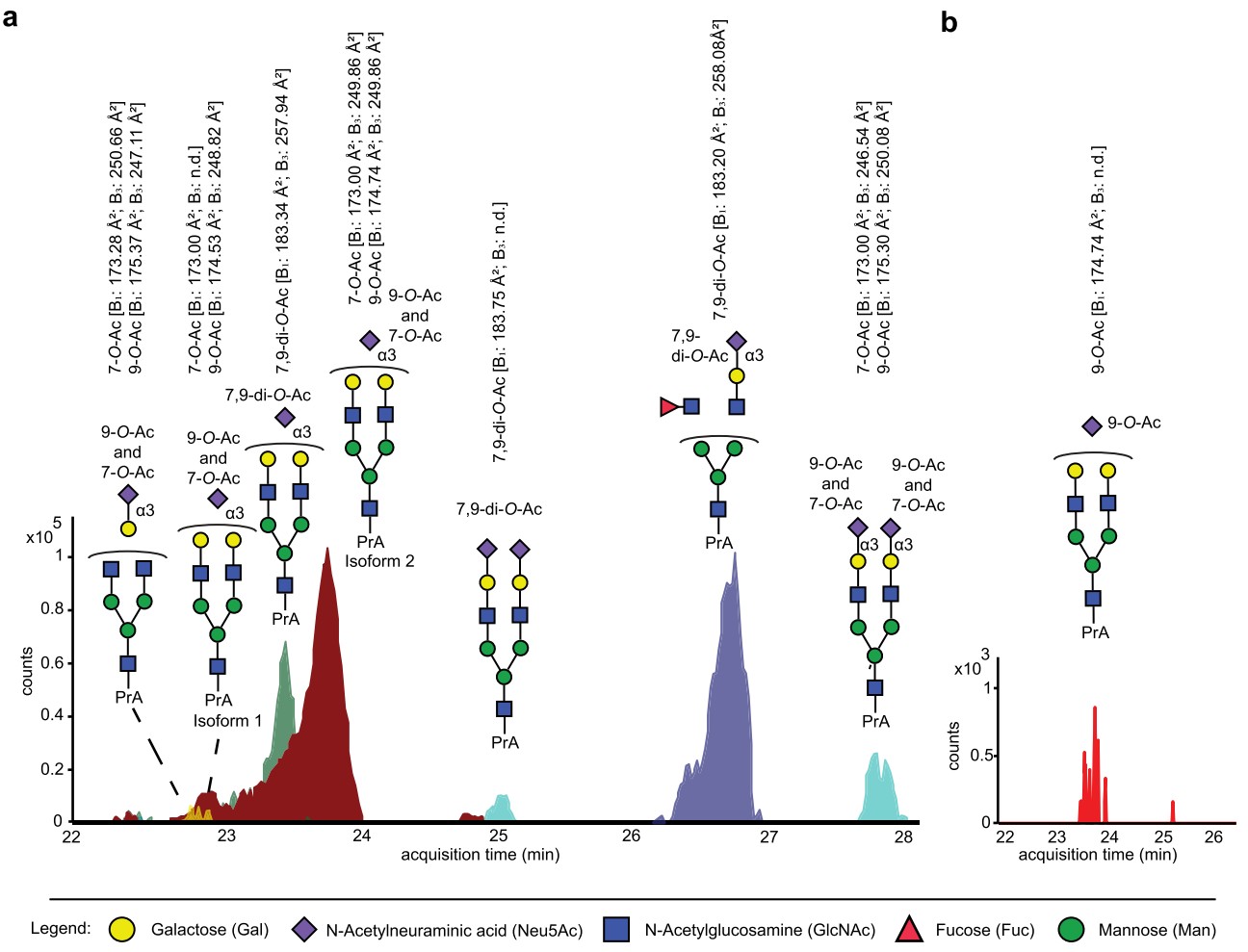

**Fig. 3 | *O*-acetylated (O-Ac) *N*-glycans derived from biologicals. a** Extracted ion chromatograms of glycans released from Myozyme and derivatized with procainamide (PrA) (*n* = 1) from left to right at *m/z* 925.8682 ([M + H + Na]²⁺, yellow), *m/z* 1006.8946 ([M + H + Na]²⁺, burgundy), *m/z* 1027.8999 ([M + H + Na]²⁺, green), *m/z* 1173.4476 ([M + H + Na]²⁺,cyan), *m/z* 1030.8934 *m/z* ([M + 2Na]²⁺, dark blue) and (**b**) Aflibercept (*n* = 1) (at *m/z* 1017.8856 ([M + 2Na]²⁺). The peaks were assigned to structures using accurate *m/z* values and sialic acid linkage and acetyl ester positions were determined by the CCS values of B₁ and B₃ ions; n.d. not determined.

sialosides were only sparsely detected (Fig. 4a, b)[54]. For monoacetylated Neu5Gc, only B₃ ions that matched with a α2,6 linkage configuration ($^{DT}CCS_{N2}$ = 245.16 Å²) were observed.

In addition to nasal, three tissue samples from three different sections of the trachea (frontal, middle and rear) from one horse were analyzed. Acetyl esters were observed on 32% of all detected *N*-glycans in the upper trachea, on 17% in the middle trachea and on 11% in the lower trachea, which indicates a decrease in *O*-acetylated sialic acids throughout the trachea (Supplementary Data 6–8). *O*-acetylation of these *N*-glycans was mainly found on Neu5Ac but minor amounts of *O*-acetylated Neu5Gc were also detected (less than 1% of total *N*-glycan contents). Like the nasal tissues, all tracheal tissues expressed exclusively 4-*O*-acetylated sialic acids that were predominantly α2,6-linked. In tracheal tissue, 41% of all detected Neu5Gc was 4-*O*-acetylated compared to 47 to 82% for Neu5Ac. Thus, the IM-MS analysis indicates that the prevalence of C-4 acetylation is dependent on sialoside type and is more abundantly present on α2,6-linked Neu5Ac.

Surprisingly, it was observed that core-fucosylation reduced modification by *O*-acetylated sialosides. The abundance of core-fucosylated structures was 8–88 times greater for non-acetylated sialosides than for *O*-acetylated sialosides, compared to their non-fucosylated counterparts. This observation suggests a possible biosynthetic bifurcation between core-fucosylation and 4-*O*-acetylation in equine tissues.

## *O*-acetylation of *N*-linked sialosides from equine α2-macroglobulin

Equine α2-macroglobulin is a serum protein that has inhibitory activity for human influenza A virus (IAV) infections[55,56], which is associated with 4-*O*-acetylation of sialic acid on *N*-glycans. It appears that 4-*O*-acetylation does not increase the affinity for hemagglutinin of IAV but confers resistance to cleavage by viral neuraminidases, and as a result can function as a decoy receptor[57]. Human A/H3N2 variants can become equine serum resistant by losing binding affinity to 4-*O*-acetylated sialic acid, indicating that 4-*O*-acetylation of sialic acid can assert selective pressure and possibly prevents cross-species transmission[58]. IM-MS analysis of *N*-glycans released from equine α2-macroglobulin revealed the presence of 50 *N*-glycans (Supplementary Data 9), of which 16 are modified by acetyl esters. Among these *O*-acetylated *N*-glycans are 3 structures that contain *O*-acetylated Neu5Gc which have not previously been detected in equine α2-macroglobulin. CCS values of the B₁ ions of *O*-acetylated Neu5Ac and Neu5Gc corresponded to the standards containing an acetyl ester at the C-4 position. Analysis of B₃ ions confirmed that both *O*-acetylated and non-acetylated Neu5Ac and Neu5Gc were almost exclusively present in the α2,6-linkage type. The detected linkage type is in agreement with previously reported analysis by nuclear magnetic resonance spectroscopy[56]. Additionally, we observed a ten-fold reduction in core-fucosylation among the *O*-acetylated *N*-glycans.

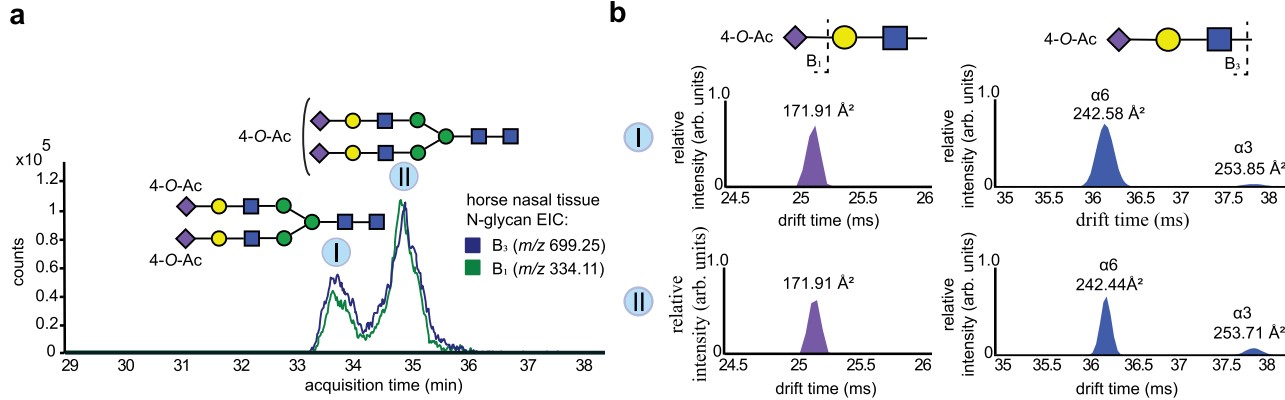

**Fig. 4 | *O*-acetylated (O-Ac) *N*-glycans derived from equine nasal tissue.**
**a** Extracted ion chromatograms (EIC) of B₃ ions of PNGaseF released glycans from one of the equine nasal tissue, analyzed by HPLC-IM-MS (*n* = 3). **b** ATDs of single charged B₁ and B₃ fragment ions of di- (I) and mono-acetylated *N*-glycans (I) released from equine nasal tissue (*n* = 1). The *O*-acetylated *N*-glycan structures were assigned using accurate mass values; Sialic acid linkage and acetyl ester positions were determined by the CCS values of B₁ and B₃ ions.

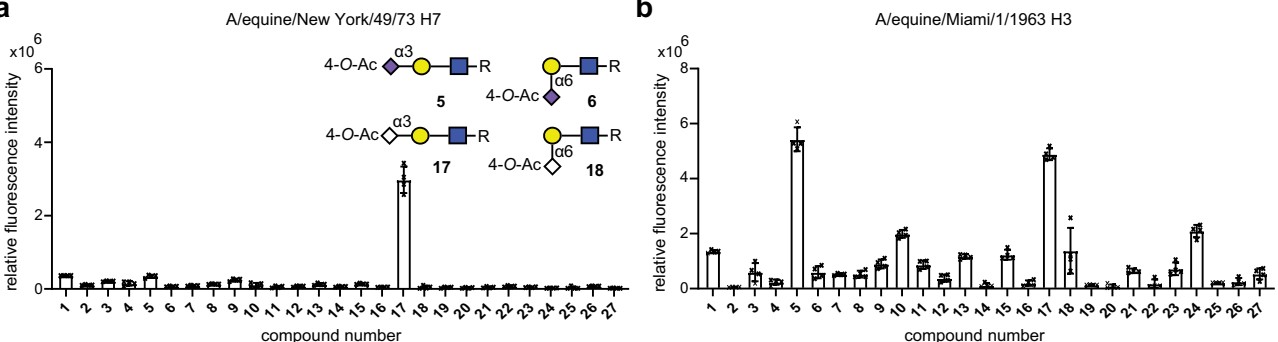

**Fig. 5 | Binding of *O*-acetylated (O-Ac) sialosides to influenza A viruses. a** Glycan microarray analysis (mean ± SD, *n* = 4) of the HA ectodomain of equine H7(A/Equine/New York/49/73 H7N7, GenBank accession no. LC414434)[59]. R = pentylaminobiotin. **b** Glycan microarray analysis (*n* = 4, with SD as error bars) of the HA ectodomain of equine H3(A/Equine/Miami/1/1963 H3N8, GenBank accession no. AAA43105.1). The compound numbers refer to the structures in Fig. 1. Source data are provided as a Source Data file.

## Glycan microarray analysis of receptor specificities of influenza A viruses

The IM-MS approach revealed that horse upper airway tissues abundantly express 4-*O*-acetylated α2,6-linked Neu5Ac receptors[57]. It is known that neuraminidases of influenza A viruses (IAVs) cannot cleave sialic acids that are modified by an acetyl ester on the *C*-4 position[57], and the utility of such a receptor may hamper viral egress. Therefore, we expect that hemagglutinins (HAs) of equine IAVs have evolved not to interact with 4-*O*-acetylated sialosides. Rather, equine IAVs appear to use the less abundant α2,3-linked-Neu5Gc as receptor[59]. The *N*-glycans of equine α2-macroglobulin are differently modified compared to those of upper airway tissues, and abundantly express α2,6-linked Neu5Ac lacking acetyl esters. Equine α2-macroglobulin is a decoy receptor for many IAVs, including those infecting humans. To gain further insight in receptor specificities of HAs and link these to receptor expression, we screened several HAs on a glycan microarray populated with the *O*-acetylated sialoglycan shown in Fig. 1b. It included the now extinct, but highly pathogenic equine A/H7N7, an equine A/H3N8 and three human A/H3N2 viruses.

The equine H7 protein showed no binding to compound **6**, which represents the most common sialoform in equine upper respiratory tissue (Fig. 5a). It was, however, observed that it can bind to 4-*O*-acetylated Neu5Gc on glycan microarray but only to the α2,3-glycosidic linkage type (compound **17**). Equine H3 showed a greater tolerance for *O*-acetylation than the examined H7 and could accommodate both Neu5Ac and Neu5Gc (Fig. 5b). This, H7 also does not bind to the most abundant sialoform expressed in equine respiratory tissue. This observation is consistent with the notion that 4-*O*-acetylated sialic acids are dead-end receptors for IAVs[57,58].

A/H3N2 was introduced in the human population in 1968 as the Hong Kong strain[57]. By glycan microarray, we evaluated the ability of this strain, and more recent H3N2 strains to bind *O*-acetylated sialosides. We selected and evaluated three time separated human A/H3N2 viruses: HK68, NL03 and CH13 (Supplementary Fig. 3 and Supplementary Tables 1 and 2). We found that strong interaction of HK68 to acetylated sialic acids is specific for 4-*O*-acetylated Neu5Ac and does not depend on linkage type. This binding ability is abolished in later strains, which suggests that the inhibition of more recent human A/H3N2 by equine α2-macroglobulin is only mediated by the non-acetylated-α2,6-linked sialic acids. The ability of 4-*O*-acetylated sialic acids detected on equine α2-macroglobulin to act as neuraminidase resistant decoy receptors appears to be lost during H3N2 evolution.

## *O*-acetylation of *O*-linked sialoglycans from mucins

Next, the scope of the IM-MS approach was extended to the analysis of acetylated *O*-glycans. Glycomics of this compound class is commonly performed by base-mediated beta-elimination to release a reducing glycan that in-situ is either reductively labeled or reduced to the

corresponding alditol[60]. The employed alkaline conditions will hydrolyze acetyl esters making the analysis of acetylated *O*-sialoglycans of *O*-glycans very challenging. Recently, we implemented an oxidative release method for *O*-glycans under neutral conditions that preserves *O*-acetylated sialoglycans. In this approach, *O*-glycans are released by hypochlorite that is neutralized to pH 6.8 to prevent degradation of glycan structures[41]. Thus, bovine submaxillary mucin (BSM) was incubated for 1 h with a 3% hypochlorite solution that was

neutralized by the addition of 1 M HCl to pH 6.8. The released glycans were isolated by solid phase extraction using porous graphitized carbon and subjected to IM-MS analysis, which identified 65 *O*-glycans (Fig. 6a and Supplementary Data 10). Sialic acids were detected on 39 glycan structures (25 Neu5Ac and 14 Neu5Gc) of which 21 were *O*-acetylated. We found 70% of all Neu5Ac to be *O*-acetylated which is higher than previously reported, although this difference can potentially be attributed to sample variations[61]. Mono-acetylated sialic acids

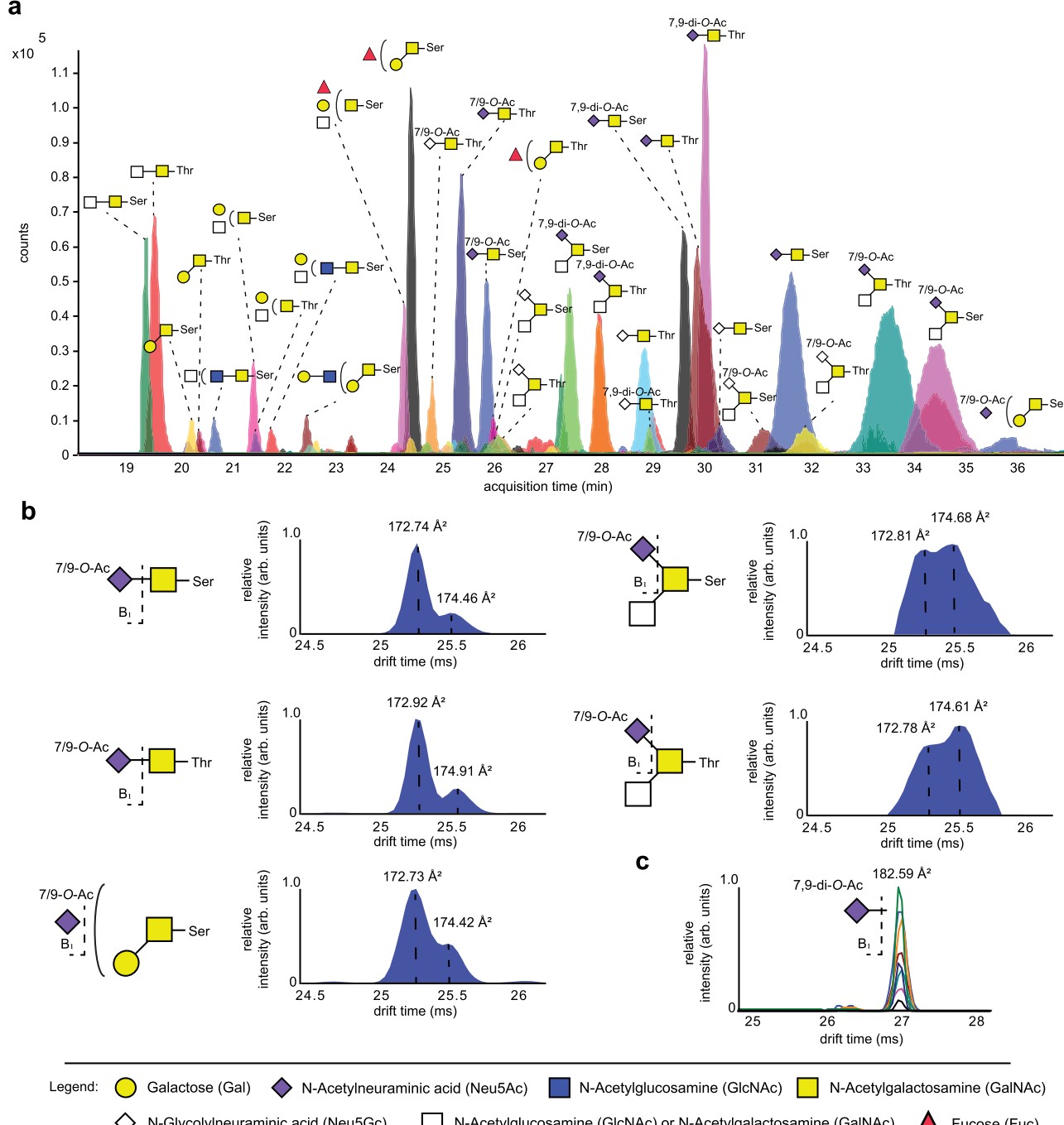

**Fig. 6 | *O*-acetylated (O-Ac) *O*-glycans derived from submaxillary mucin.**
**a** Extracted ions chromatogram and structures of the most abundant *O*-glycans (>0.1% relative concentration) of bovine submaxillary mucin (BSM) released with neutralized hypochlorite at pH 6.8 (*n* = 1). Ser and Thr refer to the conjugated amino acid in bovin submaxillary mucin. After release, the *O*-glycan previously conjugated to Ser is detected as a glycolic acid glycoside and to Thr as a lactic acid

glycoside. **b** ATDs and CCS of *O*-acetylated Neu5Ac B1 ion of selected *O*-glycan structures of BSM (*n* = 1). ATD in blue. Left dashed line corresponds to the B1 ion from the 7-*O*-acetylated Neu5Ac standard. Right dashed line corresponds to the B₁ ion from 9-*O*-acetylated Neu5Ac standard. **c** Stacked ATDs and CCS of B₁ ions from di-*O*-acetylated Neu5Ac of BSM. Dashed line corresponds to the B1 ion from the 7,9-di-*O*-acetylated Neu5Ac standard.

were detected on 14 $O$-glycans corresponding to a relative abundancy of 29%. On 17 $O$-glycans, two acetyl esters were detected which corresponds to 21% of the glycan abundancy. The oligosaccharides consist mainly of di- and trisaccharides and although no informative fragments could be identified to discriminate between isomeric cores, based on composition and the established biosynthetic pathway of $O$-glycans[62], several structures could be assigned (Supplementary Note 1). Although we were not able to differentiate between core 1 vs. core 2 and core 3 vs. 5, it is expected that the synthesis of various core structures will facilitate the development of IM-MS based methodologies for isomeric core identification.

IM-MS analysis of the $B_1$ ions from acetylated sialic acid (at $m/z$ 334.11) revealed a doublet of peaks corresponding to 7-$O$-acetyl and 9-$O$-acetyl Neu5Ac (Fig. 6b). The high abundance of 7-$O$-acetylated Neu5Ac compared to the C-9 isomer, such as STn antigen, is in agreement with previous findings that sialic acids in BSM are primarily acetylated on the C-7 hydroxyl, while 9-$O$-acetylation may be acquired via acetyl ester migration[63]. We found differences in the ratio of 7/9-acetylation among different sialylated structures, which can be due to differences in rate of acetyl ester migration or differences in preference of sialyl transferases for $O$-acetylated donors. $O$-glycans biosynthesized by ST6GalNAc-I (STn antigen), had more 7-$O$-acetylation compared to $O$-glycans modified by ST6GalNAc-II or IV (sialyl core-3 or 5). Additionally, sialyl core 3 or 5 structures were largely mono-acetylated while STn was abundantly non- and di-$O$-acetylated. No 4-$O$-acetyl $B_1$ fragments were observed, which was expected since the 4-sialate-$O$-acetyltransferase has not been identified in bovine, and acetylation on the C-4 hydroxyl cannot be achieved through migration[63]. The $B_1$ ion of di-$O$-acetylated Neu5Ac corresponded exclusively to acetylation of the C-7 and C-9 hydroxyls (Fig. 6c).

## Discussion

Understanding structures and functions of glycans is hampered by methods that can precisely determine all aspects of regio- and stereochemistry[35]. MS approaches can generally provide glycan compositions but not exact structures. Collisional- and photoactivation can result in glycosidic linkage or cross-ring fragmentation, which can give positional and regio-isomeric information. Although these approaches can provide snippets of structural information, it usually does not reveal exact structures[35].

IMS is an emerging technology that has the potential to identify isomeric glycan structures. It is the expectation that comparing experimentally derived CCS values to database values[34,64] will make it possible to determine exact structures of glycans[65]. Despite the promise of IM-MS for sequence determination of glycans, it is hampered by a lack of methods to obtain CCS values of informative glycan fragments in particular for sensitive glycan epitopes such as $O$-acetylated sialosides. Here, we addressed this challenge by taking advantage of advances in methods for chemoenzymatic synthesis of complex glycans that can provide panels of well-defined synthetic glycan standards including $O$-acetylated sialoglycans[21]. The latter type of compound was employed to create a database of $m/z$ and CCS values of $B_1$ and $B_3$ fragment ions for $O$-acetylated Neu5Ac and Neu5Gc derivatives containing 4-$O$-acetyl, 7-$O$-acetyl or 9-$O$-acetyl and combinations thereof. The resulting CCS values could be employed to assign both $O$-acetylation position as well as sialic acid linkage type of $N$-glycans and $O$-glycans of complex biological samples. The chemoenzymatic synthesis of complex glycans has progressed considerably[66], and it is now possible to prepare panels of isomeric $N$- and $O$-glycans that differ in linkage patterns. We anticipate that such compounds will be make it possible to identify CCS values for other informative fragment ions for complete sequence determination of complex glycans.

The IM-MS approach described here was showcased by analyzing $N$-glycans of two glycoprotein therapeutics. In this respect, glycans are important determinants of biological and pharmacokinetic properties of biologicals, and thus glycan structure is a critical quality attribute that must be monitored during drug development and manufacturing[67]. The methodology described here will for example make it possible to examine the influence of host cell type and culturing conditions on the expression of $O$-acetylated sialoglycans and establish their possible influence on biological and pharmacokinetic properties.

$O$-acetylated sialosides have been implicated in host-virus interactions, and the expression of specific glycoforms are expected to be determinants of host range and tissue tropism[52]. Here, we analyzed $N$-glycans of equine respiratory tissue and showed the abundant presence of C-4 acetylation on α2,6-linked sialosides. Using glycan microarray technology, we demonstrated that equine H3 and H7 IAVs can interact with 4-$O$-acetylated sialic acids but not of the α2,6-linkage type expressed in equine upper airway tissue, which is consistent with the notion that 4-$O$-acetylated sialic acids are dead-end receptors for IAVs[57,58]. It is the expectation that the use of IM-MS, to examine which $O$-acetylated sialosides are expressed by various tissues of different species, combined with glycan microarray technology to determine receptor specificities of viruses, will provide opportunities to establish to what extent differences in sialoglycan repertoire between vertebrate species hamper cross-species transmission and how zoonotic viruses have overcome these barriers.

$O$-acetylation is a post-synthetic modification that takes place in the Golgi involving sialic acid $O$-acetyltransferases (SOATs). One mammalian SOAT (CASD1, capsule structure1 domain containing 1) has been identified so far that can transfer an acetyl ester from acetyl-CoA to C-9 of cytidine-monophosphate-linked sialic acid (CMP-Neu5Ac) to give CMP-Neu5,9Ac$_2$[68]. The resulting $O$-acetylated CMP-Neu5Ac derivative can then be employed by sialyl transferases for the biosynthesis of $O$-acetylated sialoglycans. Although CASD1 is ubiquitously expressed by mammalian cell lines[69], only a limited number express cell surface $O$-acetylated sialosides. This observation has been rationalized by cell specific expression of sialyl transferase isoenzymes that have different preferences for CMP-Neu5,9Ac$_2$. The ability of IM-MS to resolve $O$-acetylation patterns as well as glycosidic linkage type provides opportunities to pinpoint possible preferences of sialyl transferases to introduce $O$-acetylated sialoglycans. For example, we observed that $O$-acetylated sialic acids displayed on $N$-glycans of equine upper airway tissues are predominantly of the α2,6-linked type. ST6Gal1 and ST3Gal4 are the enzymes that introduce 2,6- and 2,3-linked sialosides on $N$-glycans, respectively, and thus it appears that equine ST6Gal1 prefers C-4 acetylated CMP-Neu5Ac. It was also found that Neu5Ac was more prominently modified by acetyl esters compared to Neu5Gc. CMP-Neu5Gc is biosynthesized in the cytosol from CMP-Neu5Ac by CMP-Neu5Ac hydroxylase (CMAH)-catalyzed oxidation[70], and thus it appears there is an interplay between oxidation and acetylation of CMP-Neu5Ac. In the case of BSM, it was found that sialyl core 3 or 5 was largely mono-acetylated while STn was abundantly non- and di-$O$-acetylated, which suggests that the various bovine sialyl transferases have different preferences for the acetylated CMP-Neu5NAc derivatives. It is the expectation that the analysis of $O$- and $N$-glycans from various tissues of different species will provide insight in how differences in substrate specificities of isoforms of sialyl transferases may contribute to species-specific biosynthesis of sialoglycan repertoires.

## Methods

### Samples, materials, and reagents

Aflibercept was purchased from Bayer (Leverkusen, Germany). Myozyme, Influenza hemagglutinin, Endoglycosidase F2 (Endo-F2), Peptide:N-glycosidase F (PNGaseF) and recombinant hemagglutinin (HA) were produced in-house. Vivaspin2 10 kDa spinfilters were obtained from Sartorius (Göttingen, Germany). Sodium hypochlorite 15% was acquired from Acros Organics. Trifluoroacetic acid

(TFA), ethanol, dimethyl sulfoxide (DMSO), dichloromethane (DCM) and hydrochloric acid (HCl) were purchased from Merck (Darmstadt, Germany). Sodium acetate, sodium cyanoborohydride, tris(hydroxymethyl)aminomethane (TRIS), procainamide, acetic acid (LC-MS grade), formic acid (LC-MS grade), ammonium acetate, phosphate-buffered saline (PBS), PBS with Tween 20 (PBS-T) and bovine submaxillary mucin (BSM) were acquired from Sigma-Aldrich (Saint Louis, MO). Hypercarb 25 mg porous graphitized carbon (PGC) solid phase extraction (SPE) cartridges, mouse anti-streptag-Alexa647 was produced in house and goat-anti mouse-Alexa647 was obtained from Thermo Fisher Scientific (Waltham, MA, Ref#A21235, Lot#2306581). Acetonitrile (LC-MS grade) was purchased from Biosolve B.V. (Valkenswaard, The Netherlands). Ultrapure water was produced by a Synergy UV water purification system from Merck Millipore (Burlington, MA). Equine respiratory tissues and plasma were obtained from a dead horse which was sent for diagnostic and educational purposes to the veterinary pathologic diagnostic center (Faculty of Veterinary Medicine, Utrecht University, The Netherlands). α2-Macroglobulin was extracted from equine plasma. No animals were killed for this study. LC/MS calibration standard for ESI-TOF MS was obtained from Agilent Technologies (Santa Clara, CA). Streptavidin-coated glass slides were purchased from ArrayIt (Sunnyvale, CA). Three hundred KDa Spectra/Por Biotech CE Dialysis membranes were obtained from Repligen (Waltham, MA).

### Release of *N*-glycans from biologicals
*N*-glycans were released enzymatically from the proteins Aflibercept and Myozyme by Endo-F2. Aflibercept/Myozyme solutions (1 mg protein) were loaded on a 10 kDa spinfilter and centrifuged till near dryness at $2717 \times g$. A $4 \times 1$-ml volume of 100 mM sodium acetate (pH 4.5) was added and the sample was centrifuged again ($2717 \times g$) leaving at least 100 μl solution. The samples were diluted to 2 mg/ml with 100 mM sodium acetate (pH 4.5) and 10 μl EndoF2 (1 mg/ml) was added. The sample was incubated at 37 °C for 16 h under mild agitation and then purified by PGC SPE.

### Release of *N*-glycans from equine upper airway tissue
*N*-glycans were released enzymatically from equine upper airway tissue by PNGaseF. Nasal epithelium/tracheal tissue (~1 cm³) in ethanol was decanted and washed with $2 \times 20$ ml water for 15 min. To each tissue 1 ml 100 mM TRIS buffer (pH 7) and 10 μl PNGaseF (10 mg/ml) were added. Tissues were incubated for 72 h at 37 °C under mild agitation, then the samples were centrifuged for 15 min at $2717 \times g$ and the supernatant was collected, lyophilized, dissolved in a minimal volume of water and purified by PGC SPE.

### Release of *N*-glycans from equine α2-macroglobulin
α2-Macroglobulin was extracted by loading 10 ml equine plasma on a 300 KDa dialysis membrane and dialyzed at 4 °C against 5 l of ultrapure water for 24 h, with the water changed after 1 h. The remaining solution in the membrane was collected and lyophilized to yield a white powder[71]. *N*-glycans were released from equine α2-macroglobulin by PNGaseF. Equine α2-macroglobulin (1 mg) was dissolved in 1 ml 100 mM TRIS buffer (pH 7) and 10 μl PNGaseF (10 mg/ml) was added. The reaction was incubated for 24 h at 37 °C under mild agitation and then purified by PGC SPE.

### PGC SPE purification and derivatization of released *N*-glycans
The released *N*-glycans were purified by PGC SPE by equilibrating a cartridge with 1 ml of 0.1% TFA and loading the dissolved glycans on the cartridge. The cartridge was washed with 1 ml of 0.05% TFA followed by 1 ml 5%/95% ACN/water. Glycans were eluted with 50/50% ACN/water and then the eluent was evaporated under a nitrogen flow, yielding *N*-glycans for derivatization.

The positively chargeable mass label procainamide was attached to *N*-glycan free reducing ends via reductive amination. A 50-μg amount of glycan was dissolved in 120 μl water and mixed with 40 μl labeling solution (32.5 μg/μl procainamide HCl and 75 μg/μl sodium cyanoborohydride in DMSO) and 23 μl acetic acid. The mixture was vortexed and incubated at room temperature for 4 h to prevent desialylation. The reaction mixture was evaporated under nitrogen flow, redissolved in a small volume of water and extracted with $3 \times 200$ μl DCM to remove residual procainamide. Then the water layer was desalted by PGC SPE by equilibrating the cartridge with 1 ml water, loading the sample in a minimal amount of water, washing first with 1 ml of water and then with 1 ml of 5/95% ACN/water and eluting the glycans with 60/40% ACN/water with 0.1% TFA. The eluent was evaporated under nitrogen flow and analyzed by HPLC-IM-MS.

### *O*-glycan release from bovine submaxillary mucin
*O*-glycans were released from BSM using sodium hypochlorite. The pH of a 15% sodium hypochlorite solution was adjusted to 6.8 by adding ~3 ml of 1 M HCl to 5 ml of the hypochlorite solution. BSM was dissolved in water to a concentration of 1 mg/ml, 0.5 ml of the neutralized sodium hypochlorite solution (pH 6.8) was added and the reaction mixture was kept on ice for 60 min before quenching with 15 μl 1% formic acid. The mixture was directly freeze dried and then purified with PGC SPE by equilibrating the cartridge with 1 ml water, loading the sample in a minimal amount of water, washing first with 1 ml of water, then with 1 ml of 5/95% ACN/water and eluting the glycans with 60/40% ACN/water with 0.1% TFA. The eluent was evaporated under nitrogen flow and the released glycans were analyzed with HPLC-IM-MS.

### HPLC-IM-MS analysis of glycans
All standards and released glycans were analyzed with HPLC-IM-MS using an Agilent Technologies 1290 LC system coupled via a dual-source AJS electrospray interface to an Agilent Technologies 6560B drift tube ion mobility/QTOF MS instrument.

Synthetic standards were analyzed using a SeQuant ZIC-HILIC LC pre-column (Merck, Darmstadt, Germany). Solutions of standards in 80%/20% ACN/water were injected into the chromatographic system, eluted with 60%/40% ACN/water containing 0.1% formic acid at a flow rate of 0.2 ml/min and further analyzed with IM-MS.

The derivatized *N*-glycans derived from biologicals, tissue samples and equine α2-macroglobulin were dissolved in 80%/20% ACN/water, injected into the chromatographic system and separated using a ZIC-HILIC ($150 \times 4.6$ mm, 3.5 μm) column (Merck, Darmstadt, Germany), with a linear 30 min gradient from 80%/20% ACN/water containing 0.1% formic acid to 50/50% ACN/water at a flow rate of 0.25 ml/min.

Released *O*-glycans were separated on a ZIC-HILIC column ($150 \times 2.1$ mm, 3 μm particles) using 85/15% ACN/water with 0.1% formic acid for 5 min, followed by a linear gradient to 50/50% ACN/water with 0.1% formic acid over 30 min and then further analyzed by IM-MS.

IM-MS was performed with a transfer capillary voltage of 3500 V, nozzle voltage of 2000 V, nebulizer pressure of 40 psi, nitrogen drying gas at a temperature of 300 °C and a flow rate of 8 l/min and a sheath gas at 300 °C with a flow rate of 11 l/min. IM was operated with a transient rate of 16 transients/frame, a trap fill time of 3900 μs, a trap release time of 250 μs, a drift tube entrance voltage of 1400 V and a multiplexing pulsing sequence length of 4 bit. The instrument was modified with a lens that was installed at the exit of the ion transfer capillary. By increasing the voltage applied to the fragmentor, this lens creates a potential difference between the capillary and the front high-pressure funnel in the instrument. The potential difference between the fragmentor voltage and the voltage at the capillary exit determines the voltage for in-source collisionally activated dissociation. The capillary exit and high-pressure funnel entrance are kept at 360 V, resulting in a maximum potential difference of 240 V that can be

applied for ion activation when the fragmentor is set to the maximum of 600 V.

## IM-MS data processing

Masses of raw 4 bit multiplexed IM-MS data were recalibrated on reference masses with *m/z* 121 and *m/z* 922 using the IM-MS Data File Reprocessing Utility in the Agilent Technologies Masshunter software (v10.0). Reprocessed data was demultiplexed using the PNNL Pre-processor software v4.0 (Pacific Northwest National Laboratory, Richland, WA) using an interpolation of 3 drift bins and a 5 point moving average smoothing[42]. Features were identified with the Agilent Technologies Masshunter IM browser software v10.0 using an unbiased isotope model, allowing for single features with a maximum charge state of 5 and a minimal ion intensity of 500. High resolution ATDs were obtained using Agilent Technologies HRdm v2.0 software at processing level high, with an *m/z* width multiplier of 12, saturation check of 0.40 and an IF multiplier of 1.125 with SSS and Post QC enabled[43]. CCS values were calculated directly from the arrival times, by using single field CCS calibration with standards for ESI TOF-MS calibration with known *m/z* and CCS values.

## Microarray analysis

Biotinylated glycans **1**–**27** were printed in replicates of six using a Scienion sciFLEXARRAYER S3 (Berlin, Germany) on streptavidin-coated glass slides[21]. Recombinant HA was premixed with mouse anti-streptag-Alexa647 and goat-anti mouse-Alexa647 antibodies in a molar ratio of 4:2:1 in PBS-T (50 µl) for 15 min on ice. Sub-arrays (6 × 27 spots) were incubated with the premixed HA for 90 min in a humidified chamber before washing of the whole slide in 4 successive steps with PBS-T, PBS and deionized water with 5-min soak times. Arrays were dried by centrifugation and fluorescence was scanned using an Innopsys InnoScan 710 microarray scanner (Carbonne, France). The data were processed with GenePix Pro 7 (Molecular Devices, San Jose, CA) and analyzed with an in-house developed Excel macro using Excel 2016[72]. The highest and lowest replicates were removed, and the mean and standard deviation were calculated ($n = 4$). Data were plotted using GraphPad Prism 7.0 (San Diego, CA).

## Reporting summary

Further information on research design is available in the Nature Portfolio Reporting Summary linked to this article.

# Data availability

The authors declare that the data supporting the findings of this study are available within the paper and its Supplementary Information files. The ion mobility-mass spectrometric source data that support the findings of this study are available in MassIVE with the dataset identifier MSV000090864. Glycan microarray Source data are provided as a Source Data file with this paper. The Excel macro for batch processing of the glycan microarray data is available on the GitHub platform (https://github.com/enthalpyliu/carbohydrate-microarray-processing). The sequence of HA ectodomain of equine H3 (A/Equine/Miami/1/1963 H3N8) is available at GenBank, accession no. AAA43105.1. Source data are provided with this paper.

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

## Acknowledgements

We thank E. Mastrobattista and E.M. Redout for providing Aflibercept and Myozyme. This research was supported by an Agilent Technologies Applications and Core Technology University Research grant (ACT-UR-4725 to J.S.T.), the European Commission (grants 101020769 and 802780 to G.J.B. and R.P.V., respectively), Health~Holland (TKI-LSHM21030 to G.J.B.), and the Mizutani Foundation for Glycoscience (to R.P.V.).

## Author contributions

G.M.V., K.C.H., Z.L., and R.P.V. performed the experiments. G.M.V., K.C.H., R.P.V., J.F., and C.K. analyzed the data and G.M.V., K.C.H., and R.P.V. interpreted the results. G.M.V., J.S.T., and G.J.B. wrote the manuscript. J.S.T. and G.J.B. supervised the research project. All authors have given approval to the final version of the manuscript.

## Competing interests

J.F. and C.K. are employees of Agilent Technologies. The remaining authors declare no competing interests.
