## [Peer Review File · Nature Communications]

Reviewers' Comments:

Reviewer #1:

Remarks to the Author:

In the manuscript "Sialic Acid O-Acetylation Patterns and Glycosidic Linkage Type Determination by Ion Mobility-Mass Spectrometry", the authors describe a method for characterizing the sialic acids and their O-acetylation features decorating N- and O-glycans. The strength of the work is the wide library of standards that were synthesized according to a synthetic strategy they previously reported in Nature Chemistry (Li et al. 2021). The library consists in 27 glycan derivatives of Neu5Ac or Neu5Gc O-acetylated at all possible positions, connected to a galactose and an N-acetyl glucosamine, itself attached to a pentylaminobiotin group. Upon activation, these standards produce the typical glycan fragments B1 and B3, corresponding to the acetylated sialic acid and to the trisaccharide fragment after loss of the pentylaminobiotin group, respectively. The mass of these fragments is not informative of the acetylation position or of the sialic acid linkage. Here, the authors show how they can nevertheless be used for structural determination, using ion mobility spectrometry. The collision cross-sections (CCSs) of the B1 and B3 fragments of the 27 standards were measured using an Agilent drift tube IMS instrument. Then, this database is used to assign the O-acetylation features and sialic acid linkages in several biological samples, including N-glycans from Myozyme, Aflibercept2, equine α 2-macroglobulin, horse nasal tissue, horse tracheal tissue and O-glycans from bovine submaxillary mucin. The results obtained are put in relation to the binding efficiency of viruses such as the equine H7 influenza A and the host-virus interactions, which are supported with glycan microarray experiments.

One of the bottlenecks of glycan analysis using the traditional LC-MS/MS workflows is the lack of glycan standards, and the structural assignment based on the fragmentation patterns can be tedious. Combining efficient synthetic routes and ion mobility to build a data base of intrinsic CCS values has a great potential for simplifying glycan structural characterization. The work is thus of high significance for the glycomics field. However, it would benefit from a deeper investigation of the observations and a better assignment of the glycan structures. The authors are able to determine the O-acetylation features and sialic acid linkages by ion mobility, but the methods presented do not allow for sequencing the rest of the N- and O-glycans structures. The structures presented in supplementary information cannot be obtained without using fragmentation for the determination of glycans connectivity and branching. The authors claim that they can identify the O-glycan core structures based on the MS level, while this is only possible by performing MS/MS experiments. In addition, fucosylation features of N-glycans, such as antennary and core fucosylation cannot be distinguished from the m/z values, too. This puts into question the further interpretation of the results and suggestions made by the authors on the biosynthetic pathways of O-acetylation on fucosylated species.

On another hand, the application to biological samples combined with the glycan microarray analysis and the discussion on the host-virus interactions is very interesting. The conclusions are however based on the analysis of tissues coming from one individual, and the work would benefit from better statistics if this direction should be followed.

In general, the article seems to be pulled in between these two directions: presentation of an analytical workflow or investigation of the biosynthetic routes and host-virus interactions. Both are very relevant, but both also seem under-exploited.

To resume, the novelty of the work stands in the use of the very valuable standard library, whose synthetic method has recently been published by some of the authors. The figures are very clear, and the manuscript is well written. The experimental methods are described comprehensively; however they do not seem sufficient to fully characterize the structures that are presented. As it is, I therefore do not find this work sufficient for publication in Nature Communications. In the current form the article might be better suited for the scope of an analytical journal, however, with substantial revisions it may become suitable.

Detailed points

1) Some of the standards give rise to several mobility peaks (Figure 2B, structures 19 to 24), which are not commented on, and no explanation is given for their presence. This must be

addressed.

2) CCS values for the biological samples are presented in tables but the mobility profiles are not systematically compared. This would allow showing that the mobility profiles fit with the ones of the standards and that there are no other isomers present. Comparison of the broadness of the peaks could help for evaluation of the isomerism of the samples. This could be presented in FWHM included in the tables, for example.

3) The N- and especially O-glycan structures reported cannot be characterized sufficiently based on the IM-MS experiments. The authors should either provide the MS/MS data and experimental conditions if they have them, or restrict their manuscript to the O-acetylation connectivity and sialic acid linkage determination, which are already highly valuable. In the later case, they could indicate the glycan composition instead of "suggested structure" in their table.

4) In the paragraph "O-acetylation of N-linked sialosides derived from biologicals", the authors do not explain their choice of using ENDO-F2 and considering only the biantennary complex N-glycans. It would be interesting as well to know about the proportion of acetylated sialic acids in the whole sample.

5) The authors mention that such O-acetylation analysis previously required a "combination of nuclear magnetic resonance and chromatography of hydrolyzed sialic acids". It would be interesting to say few words about the consistency of the data provided by their new method compared to the previous one.

Reviewer #2:

Remarks to the Author:

The authors describe the use of ion mobility spectrometry-mass spectrometry to determine glycosidic linkages and patterns in O-acetylation present in sialic acid-containing molecules. From their results using standards, they then applied their method to determine O-acetylation in various biologically-relevant systems. Overall, this manuscript was well written with data correctly interpreted. This manuscript will likely be of interest to the readership of Nature Communications; however, there were several important points of discussion that were missing in this submission that need to be addressed prior to publication (see comments below).

The authors should better describe ion mobility spectrometry-mass spectrometry in the introduction for readers that are unfamiliar with this technique. They should define mobility (size, shape, and charge). When first describing CCS, they should emphasize these are rotationally averaged ion-neutral collision cross sections.

There is no mention of how CCS can be calculated from DTIMS-MS and also no mention of the Mason-Schamp equation. Did they use single field or multiple field to calculate CCS values?

The authors should stay consistent in using either IM-MS or IMS-MS. For example, the introduction uses IM-MS but the discussion uses IMS-MS. My preference would be IMS-MS.

The authors did a poor job of citing relevant IMS-MS publications for glycan separations. There have been papers published using SLIM, cyclic IMS, TWIMS, and TIMS related to glycan separations with ion mobility.

In the first paragraph of the discussion section, the authors attribute glycans having different CCS values because of their different conformational properties. This is confusing. While a single glycan isomer can certainly have different conformations, the primary difference in the isomers analyzed in this study are their glycosidic linkages. This portion should be re-written to say that glycans have different CCS values because of their differing structures (e.g., glycosidic linkages, anomericity, etc.).

Are the authors at all concerned that the R-group (pentylaminobiotin) used could perturb the

starting structures of the glycans and thus be part of the reason for their observed CCS differences in their fragment ions? If the authors use a different R-group (or none at all), would they observe the same CCS values?

What flow rates were used for their LC?

We thank the reviewers for their thoughtful remarks and have revised the manuscript accordingly. Details of the revisions are described below.

Reviewer #1:

In the manuscript “Sialic Acid O-Acetylation Patterns and Glycosidic Linkage Type Determination by Ion Mobility-Mass Spectrometry”, the authors describe a method for characterizing the sialic acids and their O-acetylation features decorating N- and O-glycans. The strength of the work is the wide library of standards that were synthesized according to a synthetic strategy they previously reported in *Nature Chemistry* (Li et al. 2021). The library consists in 27 glycan derivatives of Neu5Ac or Neu5Gc O-acetylated at all possible positions, connected to a galactose and an N-acetyl glucosamine, itself attached to a pentylaminobiotin group. Upon activation, these standards produce the typical glycan fragments B1 and B3, corresponding to the acetylated sialic acid and to the trisaccharide fragment after loss of the pentylaminobiotin group, respectively. The mass of these fragments is not informative of the acetylation position or of the sialic acid linkage. Here, the authors show how they can nevertheless be used for structural determination, using ion mobility spectrometry. The collision cross-sections (CCSs) of the B1 and B3 fragments of the 27 standards were measured using an Agilent drift tube IMS instrument. Then, this database is used to assign the O-acetylation features and sialic acid linkages in several biological samples, including N-glycans from Myozyme, Aflibercept2, equine α 2-macroglobulin, horse nasal tissue, horse tracheal tissue and O-glycans from bovine submaxillary mucin. The results obtained are put in relation to the binding efficiency of viruses such as the equine H7 influenza A and the host-virus interactions, which are supported with glycan microarray experiments.

One of the bottlenecks of glycan analysis using the traditional LC-MS/MS workflows is the lack of glycan standards, and the structural assignment based on the fragmentation patterns can be tedious. Combining efficient synthetic routes and ion mobility to build a data base of intrinsic CCS values has a great potential for simplifying glycan structural characterization. The work is thus of high significance for the glycomics field. However, it would benefit from a deeper investigation of the observations and a better assignment of the glycan structures. The authors are able to determine the O-acetylation features and sialic acid linkages by ion mobility, but the methods presented do not allow for sequencing the rest of the N- and O-glycans structures. The structures presented in supplementary information cannot be obtained without using fragmentation for the determination of glycans connectivity and branching. The authors claim that they can identify the O-glycan core structures based on the MS level, while this is only possible by performing MS/MS experiments. In addition, fucosylation features of N-glycans, such as antennary and core fucosylation cannot be distinguished from the m/z values, too. This puts into question the further interpretation of the results and suggestions made by the authors on the biosynthetic pathways of O-acetylation on fucosylated species.

Response. We thank the reviewer for pointing out the high significance of the approach for the field of glycomics. The current library of sialosides made it possible to assemble a data base of CCS values that cannot only identify the pattern of O-acetylation but also sialic acid linkage type. It is the expectation that additional glycan standards will make it possible to reveal other structural elements of glycans such a core type for O-glycans and branching patterns for N-glycans. We have

carefully analyzed the MS/MS data which made it possible to assign several additional structural features. Furthermore, knowledge of biosynthetic pathways of glycans has also been used to link compositions with exact structures. We have carefully revised several sections of the main manuscript. The supplementary tables have also been revised accordingly when only compositions could be determined. The following sections contain important modifications; “*Core fucosylation was demonstrated by the presence of $Y_{1\alpha}$ and Y_2 fragment ions with m/z 587.3286 and m/z 790.4080 (Supplementary Fig. 1), while antenna fucosylation could be excluded by the absence of fucosyl-LacNAc fragment ions with m/z 512.1974. α -Galactosylation was identified by a $\text{Hex}_2\text{HexNAc}$ fragment (m/z 528.1923) arising in high abundance from a single cleavage resulting in a $\text{Gal}_2\text{GlcNAc B}_3$ fragment ion (Supplementary Fig. 1). No further diagnostic fragment ions were detected in the samples for exact glycan structure determination, although the identification of specific fragment ions in combination with known biosynthetic pathways allowed for the compilation of a list of glycan compositions (Supplementary information, Table IV-IX).*

“*The oligosaccharides consist mainly of di- and trisaccharides and although no informative fragments could be identified to discriminate between isomeric cores, based on composition and the established biosynthetic pathway of O-glycans, several structures could be assigned (See supporting information for details). Although we were not able to differentiate between core 1 vs. core 2 and core 3 vs. 5, it is expected that the synthesis of various core structures will facilitate the development of IM-MS based methodologies for isomeric core identification.*”

The discussion describes the limitations of MS/MS analysis and indicates that additional glycan standards will make it possible to fully sequence complex glycans: “*The resulting CCS values could be employed to assign both O-acetylation position as well as sialic acid linkage type of N-glycans and O-glycans of complex biological samples. The chemoenzymatic synthesis of complex glycans has progressed considerably and it is now possible to prepare panels of isomeric N- and O-glycans that differ in linkage patterns. We anticipate that such compounds will make it possible to identify CCS values for other informative fragment ions for complete sequence determination of complex glycans.*”

Remark. On another hand, the application to biological samples combined with the glycan microarray analysis and the discussion on the host-virus interactions is very interesting. The conclusions are, however, based on the analysis of tissues coming from one individual, and the work would benefit from better statistics if this direction should be followed.

Response. We thank the reviewer for the supporting remarks and fully agree that it is important to provide biological replicates. The revised manuscript includes the analysis of N-glycans from nasal tissue from three different horses. These tissues expressed exclusively sialosides that are O-acetylation at the C-4 position that are predominantly α 2,6-linked, which was nearly identical to the observations for the first sample. We evaluated the data using a Venn diagram (Supplementary Fig S1). In addition to nasal, three tissue samples from three different sections of the trachea (frontal, middle and rear) from one horse has been analyzed. These all showed the same O-acetylation and sialic acid linkage pattern. It was, however, found that the relative abundance of O-acetylated sialo-glycans decreased in lower airway tissues.

Remark. In general, the article seems to be pulled in between these two directions: presentation of an analytical workflow or investigation of the biosynthetic routes and host-virus interactions. Both are very relevant, but both also seem under-exploited.

Response. Additional studies have been performed to address the concern of the referee. *O*-acetylated sialic acids have been described as “*Cinderella molecules*” and despite they are omnipresent, this class of glycan has been very difficult to study. The IM-MS approach described here makes it possible to determine the pattern of acetylation as well as the anomeric linkage type of the corresponding sialoside in complex biological samples. We have performed analysis of *N*- and *O*-glycans and included samples that have very low levels of *O*-acetylation to demonstrate the scope of the approach and expect it will be integrated in glycomic workflows. We have included several application areas such as analysis of biological drugs and tissues, mucus, and glycoproteins relevant to infection biology to indicate that the approach can be used to address various biological questions and to garner broad interest. The analytical results also uncovered aspects of biosynthesis of *O*-acetylated sialosides, which is expected to stimulate further studies. The paper highlights that the combined use of glycan standards and IM-MS has the potential to fully sequence complex glycans in biological samples. Further standards are needed to realize this potential which can be achieved by current synthetic methodologies.

Detailed points:

1) Some of the standards give rise to several mobility peaks (Figure 2B, structures 19 to 24), which are not commented on, and no explanation is given for their presence. This must be addressed.

Response. We have added the following sentence and reference to provide clarification: “Most fragment ions gave rise to a unimodal ATD, while a few distributions showed more complex signals for a singular ion, which is most likely due to the presence of different gas phase conformers⁴².”

2) CCS values for the biological samples are presented in tables but the mobility profiles are not systematically compared. This would allow showing that the mobility profiles fit with the ones of the standards and that there are no other isomers present. Comparison of the broadness of the peaks could help for evaluation of the isomericity of the samples. This could be presented in FWHM included in the tables, for example.

Response. Fragment ions from the biological samples were identified by matching their drift times or CCS values in the arrival time distributions at the peak apexes with the drift times or CCS values of arrival time distributions of the standards. The CCS values of isomers were discriminative as determined by the standards. In this way, the presence of isomeric fragment ions is revealed by the presence of several peaks and their identification can be obtained through CCS values. The FWHM of the standard replicates show relative standard deviations up to 11% after high resolution demultiplexing and cannot be used to accurately demonstrate the presence of isomers.

3) The *N*- and especially *O*-glycan structures reported cannot be characterized sufficiently based on the IM-MS experiments. The authors should either provide the MS/MS data and experimental

conditions if they have them or restrict their manuscript to the O-acetylation connectivity and sialic acid linkage determination, which are already highly valuable. In the later case, they could indicate the glycan composition instead of “suggested structure” in their table.

Response. MS/MS data have been carefully analyzed which made it possible to assign a number of additional structural elements. Compositions are indicated when no diagnostic ions could be identified. Knowledge of biosynthetic pathways have been employed to assign a number of structures and the reasoning has been provided in the SI. All tables with data have been changed according to the suggestions by the referee.

4) In the paragraph “O-acetylation of N-linked sialosides derived from biologicals”, the authors do not explain their choice of using ENDO-F2 and considering only the biantennary complex N-glycans. It would be interesting as well to know about the proportion of acetylated sialic acids in the whole sample.

Response. The biologicals predominantly carry biantennary N-glycans with very low abundant O-acetylated sialic acids. To improve signal intensities and reduce heterogeneity introduced by core fucosylation, ENDO-F2 was selected which cleaves within the chitobiose core of asparagine-linked glycans thereby removing core fucose. We have modified the following sentences in the manuscript to further explain the choice for ENDO-F2: “*The recombinant glycoproteins were dialyzed to remove additives and then the N-glycans were released enzymatically by treatment with ENDO-F2 under mild acidic conditions (100 mM sodium acetate, pH 4.5) to prevent acetyl ester migration and hydrolysis. ENDO-F2 releases the abundant biantennary complex N-glycans by cleavage of the chitobiose core of asparagine-linked glycans which should improve the detection of low abundant O-acetylated structures that may otherwise not be detected due the heterogeneity introduced by core fucosylation.*”

5) The authors mention that such O-acetylation analysis previously required a “combination of nuclear magnetic resonance spectroscopy and chromatography of hydrolyzed sialic acids”. It would be interesting to say few words about the consistency of the data provided by their new method compared to the previous one.

Response. The O-acetylation patterns of the biologicals have not been determined by NMR and only compositions of the O-acetylated structures have previously been obtained by LC-MS. The O-acetylation pattern and linkage determination of sialic acids of equine α 2-macroglobulin was previously determined by NMR and we compared the results in our manuscript with this data. The following line is included: “*The detected linkage type is in agreement with previously reported analysis by nuclear magnetic resonance spectroscopy⁵⁶.*”

Reviewer #2:

The authors describe the use of ion mobility spectrometry-mass spectrometry to determine glycosidic linkages and patterns in O-acetylation present in sialic acid-containing molecules. From their results using standards, they then applied their method to determine O-acetylation in various biologically-relevant systems. Overall, this manuscript was well written with data correctly interpreted. This manuscript will likely be of interest to the readership of Nature Communications;

however, there were several important points of discussion that were missing in this submission that need to be addressed prior to publication (see comments below).

The authors should better describe ion mobility spectrometry-mass spectrometry in the introduction for readers that are unfamiliar with this technique. They should define mobility (size, shape, and charge). When first describing CCS, they should emphasize these are rotationally averaged ion-neutral collision cross sections.

Response. We thank the reviewer for pointing out that our paper is well written with data correctly interpreted and that the manuscript will be of interest to readership of *Nature Communications*. The following sentence in the introduction has been changed: “. *In IM spectrometry (IMS), gas-phase ions are separated based on their mobility through a gas-filled drift cell under the influence of an electric field. The mobility of the ions depends on their charge state and size as well as on their shape, making IMS suitable for the separation of isomers^{29,30}. In drift tube IMS, large ions experience more ion-neutral collisions and migrate through the drift cell at a lower speed than small ions, while ions with a higher charge state migrate faster than ions with a lower charge state, resulting in a distinctive arrival time distribution (ATD) at the end of the drift cell. The arrival times can be converted into rotationally averaged ion-neutral collision cross sections (CCS), as described by the fundamental Mason-Schamp equation^{29,31}. These intrinsic CCS values are related to the surface areas of the ions and provide, in combination with m/z values, molecular descriptors for reliable compound identification.*”

Remark. There is no mention of how CCS can be calculated from DTIMS-MS and also no mention of the Mason-Schamp equation. Did they use single field or multiple field to calculate CCS values?

Response. The Mason-Schamp equation is now mentioned in the introduction (see previous comment).

To clarify how the CCS scale was calibrated, the following sentence was added to the results section: “*CCS values of the ions were directly calculated from their IM arrival times by using single field CCS calibration with standards with known m/z and CCS values (Fig. 2 and Table I).*”

Furthermore, we added the following sentence to the “Samples, materials and reagents” section: “*LC/MS calibration standard for ESI-TOF MS was obtained from Agilent Technologies (Santa Clara, CA).*” The following sentence was added to the “IM-MS data processing” section: “*CCS values were calculated directly from the arrival times, by using single field CCS calibration with standards for ESI TOF-MS calibration with known m/z and CCS values.*”

Remark. The authors should stay consistent in using either IM-MS or IMS-MS. For example, the introduction uses IM-MS but the discussion uses IMS-MS. My preference would be IMS-MS.

Response. IMS-MS has been changed into IM-MS throughout the manuscript as this is the naming used by Agilent Technologies for their drift tube devices. IMS is now only used where ion mobility spectrometry is textually used without MS.

Remark. The authors did a poor job of citing relevant IMS-MS publications for glycan separations. There have been papers published using SLIM, cyclic IMS, TWIMS, and TIMS related to glycan separations with ion mobility.

Response. The following section was added to the introduction: *“Isomeric glycans can have different surface areas and therefore may exhibit distinct CCS values³²⁻³⁴. Several studies have shown that contemporary IM-MS equipment offers sufficient resolution to separate isomeric glycans and has the potential to determine exact structures. Trapped IM-MS, for example, has been applied in combination with electronic excitation dissociation to separate several glycan fragments³⁵. Drift tube and traveling wave (TW)IM-MS have been used to separate glycan conformers³⁴ and fragment ions of sialic acid isomers³⁶, respectively, to determine sialic acid linkages of released glycans. Cyclic TWIM-MS has very high resolution capabilities and has been employed to resolve anomers and open-ring forms of oligosaccharides³⁷. In addition, the TWIM technique has been used in structures for lossless ion manipulation (SLIM)-based IMS to achieve very high resolution separation of isomeric glycans³⁸ and exact glycan structure elucidation in combination with cryogenic infrared spectroscopy-MS³⁹. Despite these advances, the challenge of implementing IM-MS for exact glycan structure determination is a lack of standards to determine CCS values of diagnostic fragment ions.”* We can add additional references if deemed necessary by the reviewer.

Response In the first paragraph of the discussion section, the authors attribute glycans having different CCS values because of their different conformational properties. This is confusing. While a single glycan isomer can certainly have different conformations, the primary difference in the isomers analyzed in this study are their glycosidic linkages. This portion should be re-written to say that glycans have different CCS values because of their differing structures (e.g., glycosidic linkages, anomericity, etc.).

Response. As indicated above, we have rewritten the section; *“Isomeric glycans (e.g. monosaccharide type, connectivity and anomeric configuration) can have different surface areas and therefore may exhibit distinct CCS values³²⁻³⁴.”*

Remark. Are the authors at all concerned that the R-group (pentylaminobiotin) used could perturb the starting structures of the glycans and thus be part of the reason for their observed CCS differences in their fragment ions? If the authors use a different R-group (or none at all), would they observe the same CCS values?

Response. We have measured several fragment ions from several glycan classes with different anomeric tags after in-source activation and did not observe any changes in CCS value of the fragment ions when different anomeric tags were used for the parent compound.

Remark. What flow rates were used for their LC?

Response. We have added the flow rates to the experimental section: *“Solutions of standards in 80%/20% ACN/water were injected into the chromatographic system, eluted with 60%/40% ACN/water containing 0.1% formic acid at a flow rate of 0.2 mL/min and further analyzed with IM-MS.”*

“The derivatized N-glycans derived from biologicals, tissue samples and equine α 2-macroglobulin were dissolved in 80%/20% ACN/water, injected into the chromatographic system and separated using a ZIC-HILIC (150x4.6 mm, 3.5 μ m) column (Merck, Darmstadt, Germany), with a linear 30 min gradient from 80%/20% ACN/water containing 0.1% formic acid to 50%/50% ACN/water at a flow rate of 0.25 mL/min.”

Reviewers' Comments:

Reviewer #1:

Remarks to the Author:

I thank the authors for their responses to my comments. The authors have carefully addressed all the suggestions for revision and the article has gained in clarity and robustness. I find it now suited for publication in Nature Communications. In providing an IMS-based method for helping with the difficult characterization of glycan structural features, the manuscript will be of high interest for the glycomics field.

Reviewer #2:

Remarks to the Author:

The authors have addressed comments from both reviewers and this manuscript should now be accepted.